# The Differences of Nutrient Components in Edible and Feeding Coix Seed at Different Developmental Stages Based on a Combined Analysis of Metabolomics

**DOI:** 10.3390/molecules28093759

**Published:** 2023-04-27

**Authors:** Xiaoyan Wei, Yong Li, Shufeng Zhou, Chao Guo, Xiaolong Dong, Qishuang Li, Juan Guo, Yanan Wang, Luqi Huang

**Affiliations:** 1College of Chinese Medicinal Materials, Jilin Agricultural University, Changchun 130118, China; 2State Key Laboratory Breeding Base of Dao-di Herbs, National Resource Center for Chinese Materia Medica, China Academy of Chinese Medical Sciences, Beijing 100700, China; 3State Key Laboratory of Exploration and Utilization of Crop Gene Resources in Southwest China, Key Laboratory of Biology and Genetic Improvement of Maize in Southwest Region, Ministry of Agriculture, Maize Research Institute of Sichuan Agricultural University, Chengdu 611130, China

**Keywords:** Coix seed, metabolomics, different developmental stages, metabolite analysis

## Abstract

*Coix lachryma-jobi* L. is an excellent plant resource that has a concomitant function for medicine, foodstuff and forage in China. At present, the commonly used cultivar for both medicine and foodstuff is Xiaobaike, and the cultivar for foraging is Daheishan. However, differences in the internal composition of plants lead to the expression of different phenotypic traits. In order to comprehensively elucidate the differences in nutrient composition changes in Coix seeds, a non-targeted metabolomics method based on ultra-performance liquid chromatography quadrupole time-of-flight mass spectrometry (LC-Q-TOF-MS) was used to analyze the metabolic changes in Coix seeds at different developmental stages. An edible *Coix* relative (Xiaobaike) and a feeding *Coix* relative (Daheishan) were selected as the research subjects. In the metabolome analysis of Coix seed, 314 metabolites were identified and detected, among which organic acids, carbohydrates, lipids, nucleotides and flavonoids were the main components. As an important standard for evaluating the quality of Coix seed, seven lipids were detected, among which fatty acids included not only even-chain fatty acids, but also odd-chain fatty acids, which was the first time detecting a variety of odd-chain fatty acids in Coix seed. The analysis of the compound contents in edible and feeding-type *Coix lachryma-jobi* L. and the lipid content at the mature stage showed that, among them, arachidic acid, behenic acid, heptadecanoic acid, heneicosanoic acid and pristanic acid may be the key compounds affecting the lipid content. In addition, in the whole process of semen coicis maturation, edible and feeding *Coix* show similar trends, and changes in the third period show clear compounds in the opposite situation, suggesting that edible and feeding *Coix* not only guarantee the relative stability of species but also provide raw materials for genetic breeding. This study provides valuable information on the formation of the edible and medicinal qualities of *Coix*.

## 1. Introduction

*Coix lacryma-jobi* L. is an annual herb of the Poaceae Coix genus that originated in Southeast Asia [1,2]. The roots, leaves and seeds of *Coix* can be used as medicine [3,4,5], especially Coix seeds, which have rich nutritional value. As the king of gramineae plants, the seeds of *Coix* are widely used in medicine, food and cosmetics in Asian countries [6,7,8]. It has been reported that Coix seeds contain polysaccharides, lipids, flavonoids, phenols, proteins, vitamins and other active ingredients [9,10]. Modern pharmacological studies have shown that these active ingredients have anti-tumor, antibacterial, antiviral and regulating blood lipids, inhibiting the pancreatic protein as well as the anti-inflammatory analgesic and antioxidant role in many aspects [11]. Of which, Coix seed oil, as the main component of Kanglaite injection, has significant therapeutic effects in clinical cancer treatment [12,13,14]. The neutral lipid of Coix seeds was identified as the main component of its anticancer activity, which was mainly triglyceride, followed by glycerol monoester, glycerol diester and fatty acid hydrocarbon ester [15,16,17]. With the increasing use of Coix seeds, the planting pressure of Coix seeds has been increasing rapidly.

*Coix lachryma-jobi* L. is widely distributed in tropical and subtropical areas of China, India, Thailand, Korea and Vietnam [18,19]. Especially in Guangxi, Yunnan, Guizhou and other Provinces in China, the unique geographical environment makes Coix seed germplasm resources extremely rich and diverse [20,21]. At present, two relative species, aquatica Daheishan (DHS) and Xiaobaike (XBK), are widely cultivated and used in China. The *Coix* aquatica Daheishan is a wild seed and the first feeding-type *Coix* in China. It has the characteristics of a high yield, good quality, good resistance and easy reproduction [22,23]. The *Coix* Xiaobaike in xingren is a cultivated seed that is one of the origin centers of Coix seeds in the world. It is a geographical marker product in the land of Coix seeds in China [22,24]. It is well known that compositional differences within plants drive plants to display different phenotypic traits. The shape, size, color and hardness of the DHS and XBK fruit have shown obvious differences during development [25,26], and the nutritive value of the feeding type and the edible type were better in the immature stage and the full maturity stage, respectively. Therefore, a study of the feeding type and edible type could not only comprehensively evaluate the nutritional differences in *Coix* fruits but could also be of great significance for the breeding of high quality *Coix* varieties in the future. In addition, there are few reports on the differences in the nutritional composition of Coix seed. Therefore, we focused on the evaluation and analysis of the differences in nutritional composition between DHS and XBK Coix seeds at different developmental stages.

In recent years, high-throughput non-targeted metabolomics and statistical methods have been commonly used to analyze plant metabolites, which makes the research of crop quality and molecular breeding progress rapidly [27,28]. The objective of this study was to comprehensively interpret XBK and DHS composition changes during development, through the collection of six different development stages of the two varieties of Coix seeds, and to use the LC-QTOF-MS-based non targeted metabolomics metabolites of Coix seeds, as well as to determine the differences between the two different varieties of metabolites via qualitative and quantitative analysis. It can not only help us to better understand the composition changes in these two species during fruit development, but can also provide a research basis for breeding high-quality cultivars. In addition, the comprehensive analysis of the metabolites in Coix seed provides important reference value for gene mining in biosynthetic pathways and in the synthetic biological production of these substances.

## 2. Results and Discussion

### 2.1. Untargeted Metabolome Analysis of Fruits of Coix in Different Cultivars

The metabolites of two *Coix* cultivars at six developmental stages were detected via LC-QTOF-MS. A total of 1723 compounds were obtained from 72 *Coix* samples (Appendix A), of which 314 compounds were identified (Appendix A). The total ion current (TIC) overlap rate of the quality control samples was high (Appendix A), and the retention time and peak area stability of the internal standard (IS) were also good (Appendix A), which indicated that the instrument data acquisition stability was good and that the experimental results were reliable. In order to better understand the nutritional and medicinal changes in feeding and edible Coix seeds during development, principal component analysis (PCA) was performed on the metabolites of 72 Coix seed samples to provide information on the overall metabolic differences between each group and the variation degree between samples within the groups. As shown in Figure 1, the accumulation patterns of metabolites in *Coix* fruits of different cultivars were quite different.

To further determine the changes in metabolite accumulation patterns of Coix seeds at different developmental stages, all samples were statistically classified with a clustering heat map. Six repetitions of each Coix seed sample were clustered together, and there was little difference among all samples, which was consistent with relevant results (Appendix A). It is widely known that the components and content of compounds in plants are the key factors affecting their quality. The cluster heatmaps of all compounds showed that the chemical contents of XBK and DHS Coix seeds showed similar trends in the whole developmental stage; the chemical changes in the two cultivars were similar at stages 0–2 and 3–5. In addition, the Coix seeds of DHS gathered together in stage H4 and H5. The samples in periods S3 and S4 were clustered together, and the samples in period S5 were isolated into clusters. In addition, more compounds showed down-regulation, suggesting that the growth rate of XBK was faster than that of DHS. Interestingly, the number of up-regulated compounds in DHS in period H3 was significantly more than that in other periods, and the samples of XBK in periods S3, S4 and S5 were clustered into a large group. Changes in the compounds were more similar, so it was speculated that the collection of DHS in period H3 was the best. The XBK could be harvested in periods S3 and S4 (Appendix A). The comprehensive analysis of changes in the metabolites of Coix seeds in different periods laid a theoretical foundation for scientific and rational harvesting and reducing the loss of nutritional components.

### 2.2. Annotated Analysis of All Compounds

Statistical mapping was conducted for all identified metabolites based on the metabolite classification information provided by the HMDB database, and annotation analysis was performed for all metabolites in 72 Coix samples. As shown in Figure 2A and Appendix A, they mainly include carboxylic acids and derivatives (65), lipids (59), sugars (51), nucleotides and derivatives (42), flavonoids (19), heterocyclic compounds (16), amine compounds (14), vitamins and hormones (13), alkaloids (8), phenolic acids (9), coumarin (3), etc. In addition, KEGG enrichment analysis was performed on all identified compounds. Among them, metabolic pathways, the biosynthesis of secondary plant metabolites, central carbon metabolism in cancer, the biosynthesis of phenylpropanoids and the biosynthesis of plant hormones are the most important metabolic pathways (Figure 2B, Appendix A).

The lipid in Coix seeds is the key to evaluating the medicinal quality of Coix [29]. Seven lipids were observed from the metabolites in the annotations in the Lipidmaps database, as shown in Figure 3 and Appendix A. The fatty acids were fatty acyls (41), prenol lipids (7), polyketides (13) and sterol lipids (3), glycerophospholipids (4), sphingolipids (1) and saccharolipids (1). Fatty acids in plants are an important part of lipids, and it was observed that the fatty acids in *Coix lachryma-jobi* L. could be divided into saturated fatty acids and unsaturated fatty acids according to unsaturation. It is worth noting that the free fatty acids in Coicis seeds can be divided into even- and odd-chain fatty acids (tridecanoic acid, pentadecanoic acid, heptadecanoic acid, heneicosanoic acid and tricosanoic acid) according to the number of carbon atoms. This is the first time that multiple species of odd-chain fatty acids have been observed in Coix seeds.

### 2.3. Screening and Analysis of Differential Compounds between the Two Cultivars

In order to further understand the metabolic differences between the Coix of DHS and XBK during development, differential metabolites were screened by combining the multiple of the difference and variable importance in the projection values of the OPLS-DA model, and significant differences were found among different groups.

#### 2.3.1. Analysis of Differential Metabolites in XBK

In the pairwise comparison of development stages, the result showed all the differential metabolites of S0 vs. S1; S0 vs. S2; S0 vs. S3; S0 vs. S4; and S0 vs. S5, which included 857 DEM (788 up-regulated, 69 down-regulated), 890 DEM (796 up-regulated, 94 down-regulated), 790 DEM (610 up-regulated, 180 down-regulated), 842 DEM (671 up-regulated, 171 down-regulated) and 821 DEM (618 up-regulated, 203 down-regulated), respectively. The total number of up-regulated and down-regulated metabolites in stages 1–5 of Coix seed development was similar to that in the early stage, but the number of up-regulated metabolites in each stage was significantly more than that of the down-regulated metabolites (Figure 4, Appendix A). Among all the differential metabolites of XBK, 314 common differential compounds existed in the six periods, and 60 of them were annotated in KEGG metabolic pathways, which were mainly involved in metabolic pathways and the biosynthesis of secondary metabolites, such as the TCA cycle, the metabolism of ascorbate and aldarate, and the metabolism of C5-Branched dibasic acid (Figure 5A, Appendix A). In addition, we also performed a cluster analysis of the common differential compounds in different periods in *Coix lachryma-jobi* L. The results showed that there were significant differences in the accumulation of compounds in different periods. At the early stage of fruit growth, about 1/3 of the compounds were accumulated in a large amount, and the trend was opposite in other periods. During the maturation process of stages 1–5, about 1/3 of the compounds accumulated more in stages 2–3, and about 1/3 of the compounds accumulated more in stages 4–5. Interestingly, the compounds that accumulated a large amount in the early stage showed a downward trend in the later stage, whereas the compounds that accumulated less in the early stage showed a large amount in the later stage. It is worth noting that, in the XBK Coix, differences for fatty acids compounds accumulate more, not only containing even-chain fatty acids (arachidic acid and behenic acid) and accumulating more differential compounds at the maturity stage, but also including odd-chain fatty acids (heptadecanoic acid, pristanic acid and heneicosanoic acid) (Figure 6, Appendix A).

#### 2.3.2. Analysis of Differential Metabolites in DHS

In the pairwise comparison of developmental stages, the results showed all the differential metabolites of H0 vs. H1; H0 vs. H2; H0 vs. H3; H0 vs. H4; and H0 vs. H5, which included 444 DEM (352 up-regulated, 92 down-regulated), 890 DEM (598 up-regulated, 88 down-regulated), 765 DEM (631 up-regulated, 134 down-regulated), 805 DEM (590 up-regulated, 215 down-regulated) and 836 DEM (576 up-regulated, 260 down-regulated), respectively (Figure 4, Appendix A). The number of differential compounds in period H2-H5 was similar, whereas the number of differential compounds in period H1 was significantly less than those in other periods. Significantly, the total number of differential compounds in XBK Coix seeds was similar, which indicated that the development of DHS Coix seeds was slower. This result is consistent with the results of the heat map analysis of all compounds (Appendix A). A total of 191 compounds were changed in six time periods of Coix seed samples, among which the number of unique differential compounds was the largest (100) in time period H0 vs. H5. The results of the annotation analysis of all the differential compounds showed that 31 compounds were annotated into the KEGG metabolic pathway, which were mainly involved metabolic pathways (51, 68.92%), the biosynthesis of secondary metabolites (25, 33.78%) and the biosynthesis of amino acids (8, 10.81%) (Figure 5B, Appendix A). In addition, cluster analysis of the common differential compounds in DHS showed that there were significant differences in the accumulation of the compounds in different periods, and about 1/2 of the compounds were accumulated in the early fruit growth period, whereas the opposite trend was observed in other periods. During the maturation process of stage 1–5, about 1/2 of the compounds accumulated more in stage 1–2, and about 1/2 of the compounds accumulated more in stage 3–5. Interestingly, compounds that accumulated a lot in the early stage showed an opposite trend in the later stage, whereas compounds that accumulated less in the early stage showed more accumulation in the later stage. In the early stage of fruit development, the main differential compounds that accumulate in Coix seed are sugars (ribose, fructose, lactate, and so on) and fatty acids (erucic acid, azelaic acid). However, there are many kinds of differential compounds during the gradual maturation of Coix seeds (Figure 7, Appendix A).

### 2.4. Differenence of Lipid Contents between the Two Cultivars

Although both the edible Coix relative (XBK) and the feeding Coix relative (DHS) are rich in nutritional value, they are usually used in different fields. In order to clarify the differences in the material basis between feeding-type Coix and food-type Coix, significantly differential metabolites were screened out from the two cultivars (Figure 6, Figure 7). The fatty acid species of feeding-type Coix in late development accounted for significantly more than that of edible-type Coix, which may be the specific compounds that distinguish the two cultivars. In order to further verify the obvious differences in lipid content between the two cultivars, the lipid content of edible Coix seeds and feeding Coix seeds in the mature stage was detected, and the results showed that there were significant differences in the lipid content in the Coix seeds of the two different varieties. The lipid content of XBK Coix seeds (H) of Xingren was 1.6 times that of DHS Coix seeds (S) (Figure 8, Appendix A). These results indicate that the differential fatty acid compounds found in XBK may be the key compounds affecting lipid content (arachidic acid, behenic acid, heptadecanoic acid, pristanic acid and heneicosanoic acid).

## 3. Discussion

Coix seed have a long history of application in China, in which it has been widely used in traditional Chinese medicine [30]. At present, two varieties are widely used for food and feeding, among which Coix seeds are the most popular edible variety in Xingren, and DHS is the first foraged variety in China [23,31]. Coix seeds, as medicinal and edible organic green food, is rich in polysaccharides, lipids, flavonoids and vitamins, which contain many nutrients, among which lipids are an important standard of semen coicis quality assessment [29,32]. However, the overall differences in Coix seed oil have not had a comprehensive study until now, thus limiting Coix seed variety breeding and the development and utilization of nutrients. In this study, A systematic analysis of dietary and foraged Coix seeds was carried out via metabolomics, highlighting the Coix seed oil potential development value and revealing the influence of DHS and XBK content differences in compounds. Analyzing lipid biosynthesis pathways in Coix provides important resources. It is conducive to the cultivation of new cultivars with better quality.

In order to evaluate the nutritional composition of Coix seed from edible and foraged Coix seeds, the metabolites of Coix seeds from XBK and DHS Coix seeds were qualitatively and quantitatively analyzed via LC-QTOF-MS. In different stages of Coix seed development, XBK and DHS seeds showed similar changes in compounds, mainly showing the low characteristics in the early stage and high characteristics in the late stage (Appendix A). Generally, the composition and content of active compounds in plant fruits are affected by their ripening stage [33]. In the process of maturation, it is interesting to note that DHS samples gathered in periods H4 and H5, and XBK samples gathered together in periods S3 and S4. However, the samples from period S5 were separated into a cluster, indicating that the growth rate of XBK was faster than that of DHS. In addition, the number of up-regulated compounds in DHS at stage H3 was significantly higher than that in other stages, including that in XBK in S3 and that in samples S4 and S5, which gathered into a major category. In addition, more compounds showed a lower trend in stage S5, which indicated that the DHS in period H3 of harvesting is the best, and that the XBK harvest was better during periods S3 and S4.

Primary and secondary metabolites in plants are not only essential for plant growth and development but also have important nutritional value and are important sources of nutrients for the human body [34]. In the non-targeted metabolome analysis of Coix seed, a total of 314 metabolites were identified from all samples, among which organic acids, carbohydrates, lipids and flavonoids were the main nutrients in Coix seeds. Lipids are one of the most important nutrients that the body needs and are composed of fatty acids, glycerolipids, glycerophospholipids, sphingolipids, sterol lipids, prenol lipids, saccharolipids and polyketides [35,36]. Most of these fatty acids can be synthesized in the human body, except linoleic acid, linolenic acid and arachidonic acid, which can only be obtained from food [37]. In addition, lipids play important roles in plant growth and development. Some very-long-chain fatty acids can reduce water transpiration, improve drought tolerance, reduce UV damage and resist the effects of many plants via the synthesis of waxes [38,39]. Fatty acids are an important part of lipids. By analyzing the free fatty acids in the metabolic data of all samples and classifying them according to the odd and even numbers of carbon atoms, even-chain fatty acids were observed in the metabolites, such as palmitic acid, stearic acid, oleic acid and linoleic acid. A variety of odd-chain fatty acids were also observed, including tridecanoic acid, pentadecanoic acid, heptadecanoic acid, heneicosanoic acid and tricosanoic acid. Compared with the even-chain fatty acids, although the number and type of odd-chain fatty acid content is less in Coix, studies have found that fatty acids with an odd number of chains can also inhibit the proliferation of cancer cells and may become even-chain fatty acids as a useful alternative [40,41]. The results not only indicate the nutritional value of Coix but also suggest that odd-chain fatty acids have great potential in the development and utilization of its medicinal value.

In addition, we performed cluster analysis on differential compounds of the two *Coix* cultivars. The results show that there was arachidic acid, behenic acid, heptadecanoic acid, pristanic acid and heneicosanoic acid in the maturation stage of Coix. However, the mature compounds of DHS did not show a large number of fatty acid compounds. The analysis of compounds and the detection of lipid content in the mature stage showed that the lipid content of DHS was significantly lower than that of XBK. In summary, arachidic acid, behenic acid, heptadecanoic acid, pristanic acid and heneicosanoic acid may be the key compounds that affect their lipid content. Although the Coix seeds of lipid compounds have significant differences, studies on the treatment of various malignant tumors with Coix seed oil are mainly on unpurified oil [42], which not only leads to unknown key genes in the biosynthesis process of Coix seed oil, but also to unclear regulatory mechanisms, seriously limiting the biotechnological breeding of Coix seed oil. Therefore, the analysis of the metabolome of Coix seed oil lays the theoretical support for the subsequent biosynthesis of lipids.

## 4. Materials and Methods

### 4.1. Plant Material

The two Coix varieties were identified by the Maize Research Institute of Sichuan Agricultural University in this study. Coix was obtained from wild aquatic Coix (*C. aquatica*) in Gasa Town (100°45′43″ N, 21°57′22″ E), Jinghong City and Yunnan Province through 7 generations of selfing. Another material is Xiaobaike Coix (*C. chinensis*) from Xingren County (100°54′ N, 25°16′ E), Guizhou Province, China, which is an annual conventional plant widely grown in southern China. During the growth stage, both varieties were under the same field management conditions. Samples were collected every 7 days from fertilization until fruit ripening, and a total of 6 periods were obtained. A total of 6 periods (0, 1, 2, 3, 4 and 5) were obtained. All samples were selected for 6 biological replicates in each period (Figure 9). This is the entire period of Coix fruit ripening, and all samples were frozen in liquid nitrogen and kept in a −80 °C freezer for further use.

### 4.2. Extraction of Metabolites

Seed samples of 50 mg of Coix were taken from each sample, and a 1000 μL solution containing internal standard (1000:2) was added (the volume ratio of methanol, acetonitrile and water was 2:2:1, and the internal standard concentration was 2 mg/L), and the solution was vortically mixed for 30 s. Porcelain beads were added and treated with a 45 Hz grinding instrument for 10 min and with ultrasound for 10 min (ice water bath). Then, it was set to −20 °C for 1 h and centrifuged at 12,000 rpm for 15 min, and the supernatant was taken for drying. Finally, 160 μL of extract (volume ratio of acetonitrile and water was 1:1) was added to the dried metabolites for redissolution, which was vortexed for 30 s and put in an ice water bath ultrasound for 10 min, and samples were centrifuged at 4 °C with 13,000 rpm for 15 min. The supernatant was filtered with a 0.22 μm filter membrane, and 10 μL of each sample was mixed for detection [43,44].

### 4.3. UPLC-MS/MS Analysis

The detection methods of samples refer to previous reports [45]. The LC-MS system consisted of an ultra-high-performance liquid phase (Acquity I-Class PLUS, Waters Corp., Milford, CT, United States) and a high-resolution mass spectrometer (Xevo G2-XS QT). The chromatographic column was an Acquity UPLC HSS T3 (1.8 μm 2.1 × 100 mm) purchased from Waters (Waters Corp., Milford, CT, United States). The liquid chromatographic conditions were as follows: mobile phase A, 0.1% formic acid aqueous solution; and mobile phase B, 0.1% formic acid acetonitrile, with an injection volume of 1 μL and a flow rate of 400 μL/min. The solvent gradient was set as follows: The linear gradient was set as follows: 0–0.25 min: 98% A, 0.25–10 min: 98% A to 2% A, 10–13 min: 2% A, 13–13.1 min: 2% A to 98% A, remaining as such until 15 min. The ESI ion source parameters were as follows: capillary voltage, 2000 V(positive ion mode) or −1500 V(negative ion mode); taper hole voltage, 30 V; ion source temperature, 150 °C; dissolvent temperature, 500 °C; reverse air flow rate, 50 L/h; and desolvent gas flow rate, 800 L/h. The raw data were collected using MassLynx software (version 4.2, Waters Corp., Milford, CT, United States) and processed by the Progenesis QI software (Waters Corp., Milford, CT, United States). The identification and theoretical fragment identification were performed based on an online METLIN database of Progenesis QI software and a self-built database of BioMarker Technologies at the same time, all within 100 parts per million (BioMarker Technologies, Beijing, China).

### 4.4. Data Analysis of Metabolome

In order to guarantee the quality of the collected data, for each sample, the retention time and the peak area of the overlapping degree of stability were verified through the instrument and the internal standard. However, internal standard L-2-chlorophenylalanine was mixed into each sample for quality control, and the stability of instrument data acquisition was judged by comparing the retention time of the internal standard and the stability of the peak area. Based on MetaboAnalystR, principal component analysis and cluster heat map analysis, standardization of the metabolites data was conducted, and the pathway enrichment of differential metabolites in the samples were statistically analyzed [46]. The KEGG Database (https://www.kegg.jp/kegg/compound (accessed on 6 May 2020)) was used to investigate the annotated pathways of compounds (https://www.kegg.jp/kegg/pathway.html (accessed on 6 May 2020)), including the metabolism of carbohydrates, nucleosides, amino acids and biodegradation of organic compounds. The screening criteria of differential metabolites were log2 (fold change) > 1.50 and VIP (variable influence on projection) > 1 by BMKCloud (www.biocloud.net (accessed on 9 May 2020)) in the analysis.

### 4.5. Determination of Total Lipid in Coix Seeds

The extraction of oil in Coix seeds was in accordance with the determination method of crude fat in food issued by the Ministry of Health of the People’s Republic of China (GB/T 5009.6-2003 [47]). Dry Daheishan Coix and Xiaobaike Coix were weighed, and three biological replicates were selected for each sample. Coix seeds were crushed and passed through a 40-mesh sieve after removing the shell and seed coat. Samples weighing 2 g were put into the Soxhlet extraction tube, and petroleum ether was added for extraction. The extraction liquid was reflustered 6 times/h. After 6 h of extraction, the extracted liquid was dried until obtaining a constant weight.

## 5. Conclusions

In this study, we described the metabolomic characteristics of foraged (DHS) and edible (XBK) Coix seeds and analyzed the fatty acids that may contribute to differences in lipid content. This is the first time that the nutrient component of Coix seed oil has been studied via metabolomics. Our data not only provide basic theoretical support for breeding new varieties with high Coix oil production but also provide the theoretical basis for further understanding the biosynthetic pathway and regulatory mechanism of Coix seed oil.

## Figures and Tables

**Figure 1 molecules-28-03759-f001:**
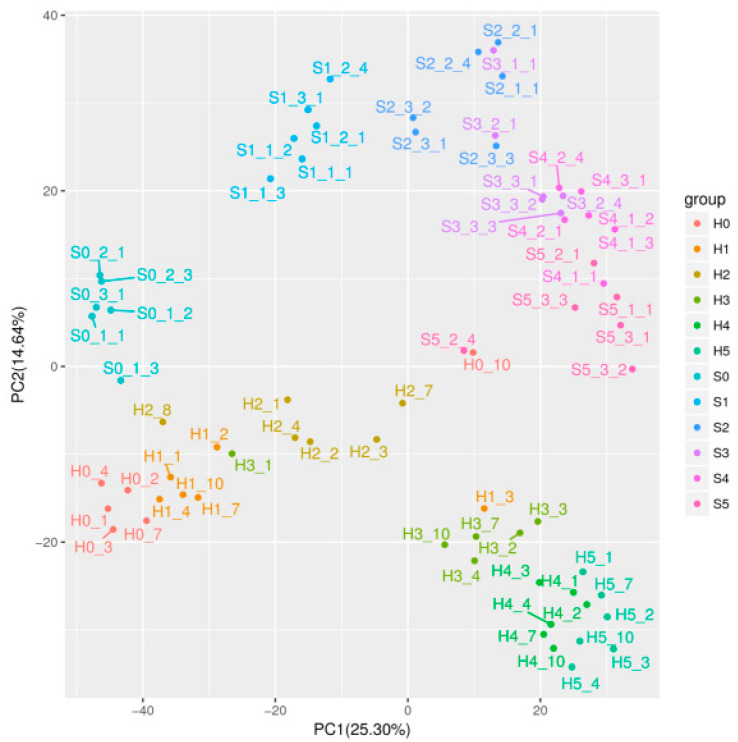
PCA scores plot for DHS and XBK Coix seeds.

**Figure 2 molecules-28-03759-f002:**
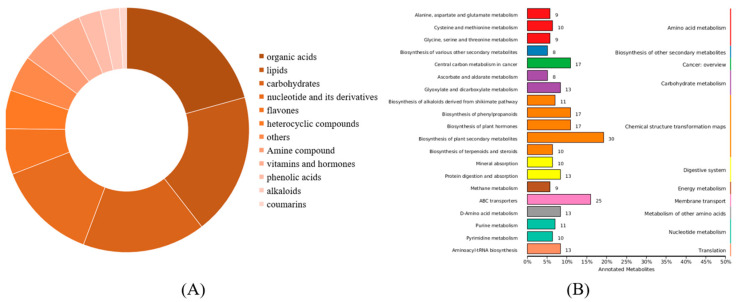
Metabolites in different cultivars of Coix seeds: (**A**) Types of the identified metabolites from DHS and XBK. (**B**) Enrichment analysis of metabolite KEGG in *Coix lachryma-jobi* L.

**Figure 3 molecules-28-03759-f003:**
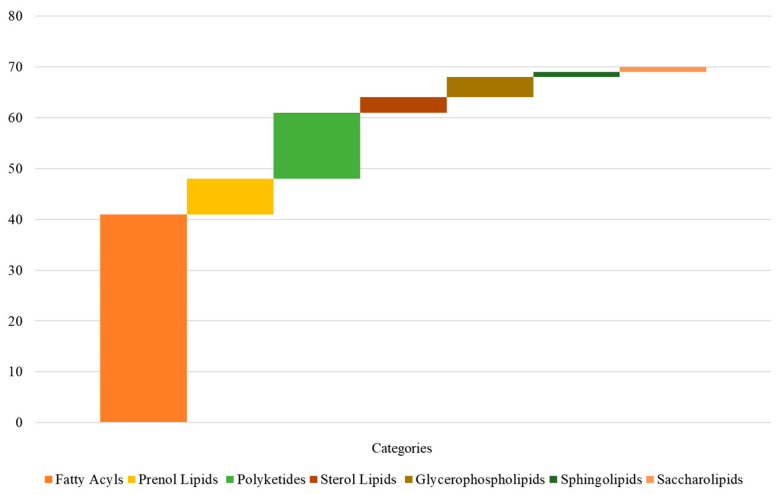
The types and quantities of identified lipids from Coix seeds.

**Figure 4 molecules-28-03759-f004:**
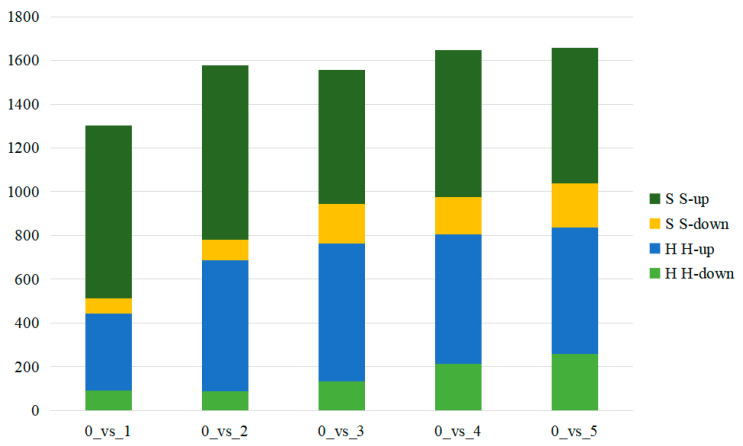
The number of differential metabolites in DHS and XBK.

**Figure 5 molecules-28-03759-f005:**
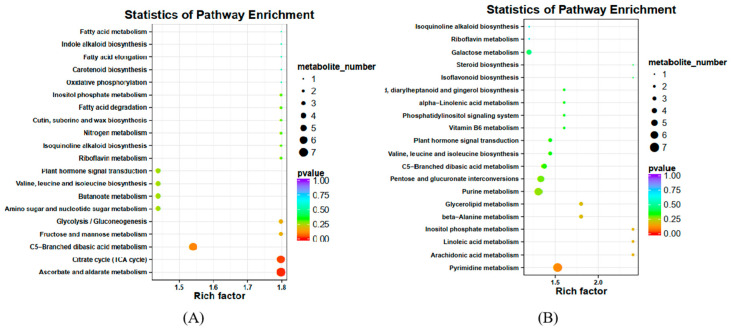
Statistics of pathway enrichment of differential metabolites in two varieties: (**A**) Statistics of pathway enrichment of differential metabolites in XBK. (**B**) Statistics of pathway enrichment of differential metabolites in DHS.

**Figure 6 molecules-28-03759-f006:**
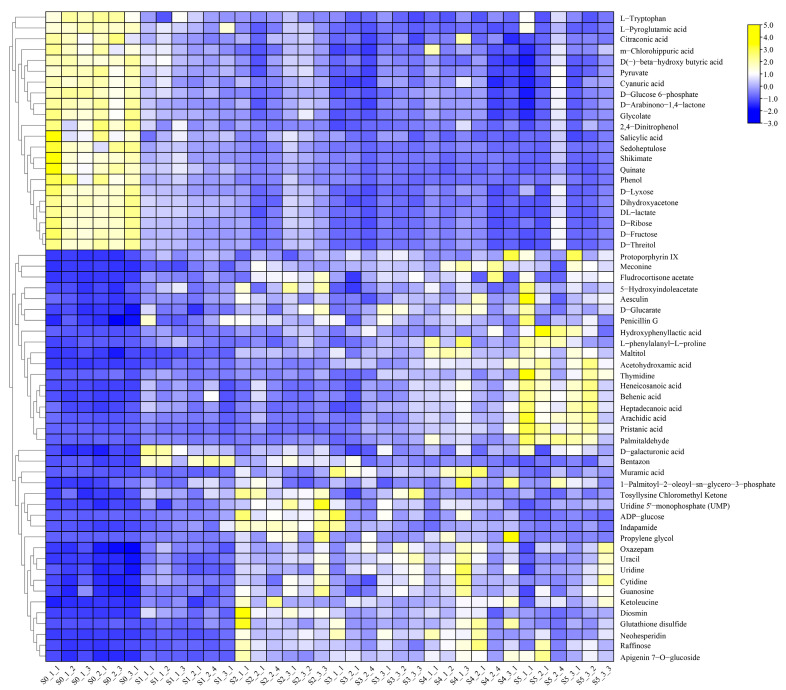
Cluster analysis of identified differential metabolites from XBK.

**Figure 7 molecules-28-03759-f007:**
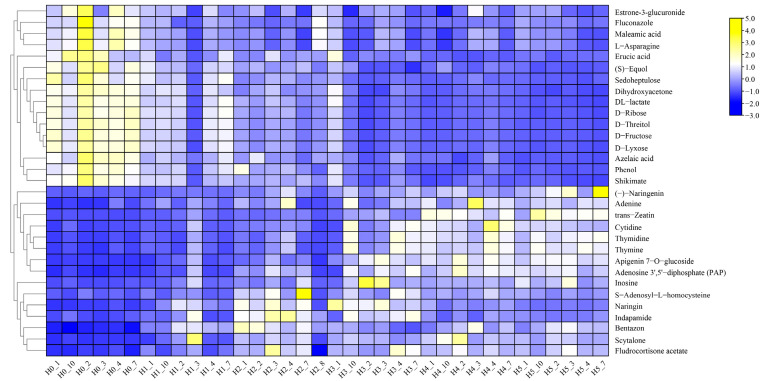
Cluster analysis of identified differential metabolites from DHS.

**Figure 8 molecules-28-03759-f008:**
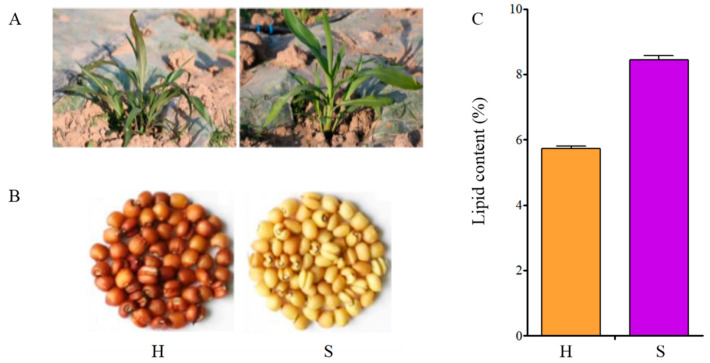
Traits of *Coix lachryma-jobi* L: (**A**) Seedlings of *Coix lachryma-jobi* L. (**B**) Seeds of *Coix lachryma-jobi* L. after hulling. (**C**) Contents of lipids in Daheishan Coix seeds (DHS) and Xiaobaike Coix seeds (XBK) (mean *±* SD, *n* = 3).

**Figure 9 molecules-28-03759-f009:**
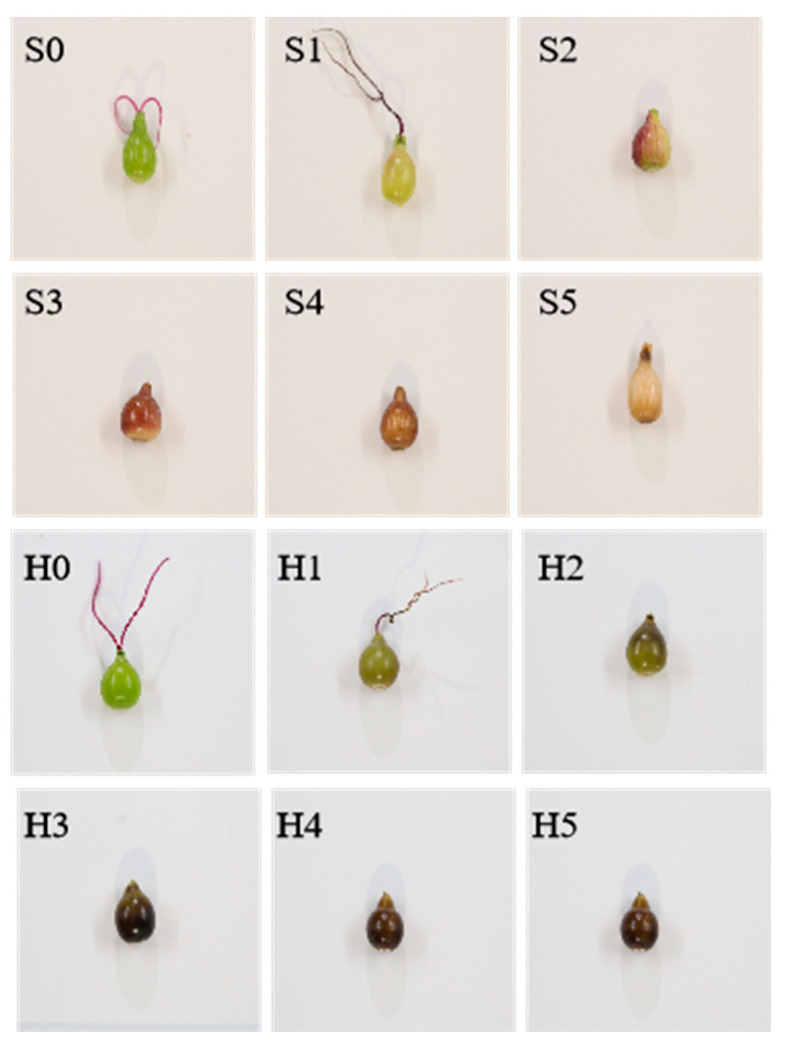
Samples of Coix seeds from different periods. (S and H stands for Daheishan and Xiaobaike Coix seeds, respectively. Scale bar = 1 cm).

## Data Availability

The data in this study are available in the article and in the Appendix A.

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
