# Peer review of "The Differences of Nutrient Components in Edible and Feeding Coix Seed at Different Developmental Stages Based on a Combined Analysis of Metabolomics"

_molecules, 2023, doi:10.3390/molecules28093759_

Round 1
Reviewer 1 Report
The subject of the research presented in the manuscript is interesting. However, the study's experimental and data analysis parts are imprecise, neglectful, and chaotic. As a result, the scientific value of the manuscript is very low.
1) The procedure to identify detected features is a farce. The authors claim to identify detected compounds by MELIN and an in-house database. Yet, on their list in Table S1, there is no data to confirm the identification accuracy. At the very least, if high-resolution MS measurements were carried out, the authors should provide the following: detected m/z, the calculated molecular formula, and an error between observed and computed m/z values. However, that only shows a match in a molecular formula, which is insufficient to identify the detected feature fully. MS/MS spectra should also be presented and compared with standards or library spectra to prove the match (in the form of, for example, cosine similarity score). Without that data, table S1 is just the authors' "wish list" on which they have various "pearls" such as, for example, "Acamprosate", which does not occur in plants naturally.
2) Quality control procedure based on a single internal standard is also insufficient because the authors analyze many compounds with different properties, from polar carbohydrates to non-polar lipids. Again, pooled QC samples should be prepared and frequently analyzed to adhere to the bare minimum of metabolomics community standards. Only features consistently detected in QC samples should be used in subsequent statistical analyses.
3) Figures in scientific manuscripts should present data. Can the authors explain what can be gathered from their heatmap in Figure 1? Not only is it unreadable, but it is also wholly unnecessary. The same is with Venn plots in Figure 5. The authors seem unfamiliar with the UpSet plot (https://en.wikipedia.org/wiki/UpSet_Plot), designed to circumvent the shortcomings of Venn if more than 4 groups are compared.
Again, I stress that presented figures must be readable. Otherwise, what's the point of showing them?
4) The results of the analysis of pathway enrichment based on metabolomics data are merely a suggestion and cannot be treated as scientific proof without the appropriate external validation. Therefore, the authors must provide chemical (direct and validated analysis of the pathway metabolites, isotopic enrichment data, etc.) or biochemical (measurements of enzyme activity, expression data, etc.) evidence that the flow of metabolites in a given pathway is increased/decreased.
5) Anyone familiar with the metaboanalyst.ca web page can easily see that at least part of the data in the manuscript was analyzed using that page or the metaboanalystR package. Yet, neither is cited.
6) Throughout the manuscript, the authors often use chemical names like D-ribose, etc. It is unlikely their chromatographic separation is chiral, allowing the separation of L and D isomers. What is, therefore, the basis of such a precise identification?
7) Before publication, the authors should extensively edit the manuscript's language. As the name suggests, Total Ion Current (TIC) differs from "total ion flow" (pg. 2, line 77).
8) Not a single chromatogram is presented throughout the manuscript. Therefore the readers don't have a chance to check if the authors were actually able to separate all these 1723 metabolites during the 15 min of chromatographic analysis
Author Response
Response to the comments of reviewer 1
The subject of the research presented in the manuscript is interesting. However, the study's experimental and data analysis parts are imprecise, neglectful, and chaotic. As a result, the scientific value of the manuscript is very low.
Resopnse: Thank you for pointing out this problem. We were ashamed that we did not make it clear in the initial manuscript. In this manuscript, we aimed to clarify the differences in the nutrient composition of Coix seed at different developmental stages. Firstly, The data detected by untargeted metabolomics were analyzed to determine the reliability of the data. On this basis, the content changes of all compounds detected in the Coix seed were analyzed, and the growth status of the two varieties as well as the best sampling time were speculated. Then, all the metabolites were annotated to identify the compounds affecting the quality of Coix seed. Secondly, the differential compounds were screened and the common differential compounds at different periods were analyzed and compared, suggesting that fatty acids may be the compounds affecting the quality of coix seeds. Finally, we determined the total oil content of the two varieties to further illustrate the above remarks.
In the following sections, we have carefully considered the reviewer's suggestions and made some changes. We hope that our response may well resolve the reviewer's questions. Thanks again to the reviewers for taking the time to review our manuscript.
Major points:
Comment 1: The procedure to identify detected features is a farce. The authors claim to identify detected compounds by MELIN and an in-house database. Yet, on their list in Table S1, there is no data to confirm the identification accuracy. At the very least, if high-resolution MS measurements were carried out, the authors should provide the following: detected m/z, the calculated molecular formula, and an error between observed and computed m/z values. However, that only shows a match in a molecular formula, which is insufficient to identify the detected feature fully. MS/MS spectra should also be presented and compared with standards or library spectra to prove the match (in the form of, for example, cosine similarity score). Without that data, table S1 is just the authors' "wish list" on which they have various "pearls" such as, for example, "Acamprosate", which does not occur in plants naturally.
Resopnse 1: Thanks for your valuable suggestion. We are very sorry for not giving the data at the beginning. We have provided the data as supplementary materials Table S1 and Table S2 and revised our manuscript accordingly. The supplementary materials include the detected m/z, the calculated molecular formula, and the error between the observed value and the calculated value.
Comment 2: Quality control procedure based on a single internal standard is also insufficient because the authors analyze many compounds with different properties, from polar carbohydrates to non-polar lipids. Again, pooled QC samples should be prepared and frequently analyzed to adhere to the bare minimum of metabolomics community standards. Only features consistently detected in QC samples should be used in subsequent statistical analyses.
Resopnse 2: Thanks for your kind suggestion. As the reviewer’s suggestion, we have added pictures Figure S1-S3 and the supplementary material Table S1 to prove that consistent features have been detected in QC samples.
Comment 3: Figures in scientific manuscripts should present data. Can the authors explain what can be gathered from their heatmap in Figure 1? Not only is it unreadable, but it is also wholly unnecessary. The same is with Venn plots in Figure 5. The authors seem unfamiliar with the UpSet plot (https://en.wikipedia.org/wiki/ UpSet_Plot), designed to circumvent the shortcomings of Venn if more than 4 groups are compared. Again, I stress that presented figures must be readable. Otherwise, what's the point of showing them?
Resopnse 3: Thanks for your valuable suggestion. The harvest date of a plant is one of the key factors affecting its quality, and the nutritional quality of plants may vary greatly according to the harvest date. Therefore, the analysis of the content changes of all compounds detected in Coix seed can lay a theoretical foundation for scientific and reasonable harvesting and reducing the loss of nutrients. Based on the reviewer's suggestions, the corresponding pictures were modified to the attached materials (Fig. S5). On the other hand, as the reviewer’s suggestion, we have removed Venn plots and added the screened compounds in the supplementary Tables S6 and S8. Thank you again for your valuable advice about UpSet plot.
Comment 4: The results of the analysis of pathway enrichment based on metabolomics data are merely a suggestion and cannot be treated as scientific proof without the appropriate external validation. Therefore, the authors must provide chemical (direct and validated analysis of the pathway metabolites, isotopic enrichment data, etc.) or biochemical (measurements of enzyme activity, expression data, etc.) evidence that the flow of metabolites in a given pathway is increased/decreased.
Resopnse 4: We appreciate it very much for this good suggestion. As the reviewer’s comment, the results of the analysis of pathway enrichment based on metabolomics data are merely a suggestion. The focus of this paper is mainly to give a high probability prediction based on the metabolomics data. We are very interested in the changes of some metabolites. As the reviewer’s suggestion, we will conduct a series of verifies of enzyme activities, expression data and functions in the target compound pathways in the future.
Comment 5: Anyone familiar with the metaboanalyst.ca web page can easily see that at least part of the data in the manuscript was analyzed using that page or the metaboanalystR package. Yet, neither is cited.
Resopnse 5: Thanks for your valuable suggestion. We are very sorry for our careless error of not referring to specific data analysis in the article. As suggested by the reviewer, we have added appropriate references and web pages at Line 393-402 for critical data analysis.
Comment 6: Throughout the manuscript, the authors often use chemical names like D-ribose, etc. It is unlikely their chromatographic separation is chiral, allowing the separation of L and D isomers. What is, therefore, the basis of such a precise identification?
Resopnse 6: Thanks for your good suggestion.The identification of the three dimensions of RT, MS1, and MS2 is the way to identify metabolites in the manuscript. In this study, the MS2 database of BioMarker (a database constructed by standards) was used for metabolite identification to obtain the name of the substance. The L and D isomers presented are the results of database comparisons. Secondary mass spectrometry matching qualitative score values and corresponding substance names have been presented in Supplementary material Table S2. Based on variation among matching qualitative score values of the isomers, combining the reviewer's suggestion and rigorism of data analysis, we have removed the detailed description of the L and D from the article at Line 233.
Comment 7: Before publication, the authors should extensively edit the manuscript's language. As the name suggests, Total Ion Current (TIC) differs from "total ion flow" (pg. 2, line 77).
Resopnse7: Thanks for your careful review. We are very sorry for our negligence of the manuscript's language. Total Ion Current (TIC) have made correction at Line 98 and the manuscript's language was check off carefully.
Comment 8: Not a single chromatogram is presented throughout the manuscript. Therefore the readers don't have a chance to check if the authors were actually able to separate all these 1723 metabolites during the 15 min of chromatographic analysis.
Resopnse8: Thanks for your careful review. I am very sorry that we overlooked the picture in the manuscript. The number of 1723 metabolites is the total number of peaks in 15 minutes, but 314 compounds can be obtained through the database comparison. Chromatographic analysis plots have been added in Figure S1 in the Supplementary material.
Reviewer 2 Report
In this study, untargeted metabolomics was used to analyze the metabolic changes of Coix seed at different developmental stages. It provides ideas for future research in the fields of agronomy and Chinese medicine. The overall design of the article is good, but it lacks the support of original data. The author is requested to upload the original data and subsequent processing data to the platform for readers to verify the authenticity and reliability of the experimental data.
Author Response
Response to the comments of reviewer 2
In this study, untargeted metabolomics was used to analyze the metabolic changes of Coix seed at different developmental stages. It provides ideas for future research in the fields of agronomy and Chinese medicine. The overall design of the article is good, but it lacks the support of original data. The author is requested to upload the original data and subsequent processing data to the platform for readers to verify the authenticity and reliability of the experimental data.
Resopnse: We would like to thank you for your careful reading, helpful comments, and constructive suggestions, which has significantly improved the presentation of our manuscript. In the revised manuscript, we have added the original data and subsequent processing data in the Supplementary material Table S1-S8. The detailed data information was provided in the supplementary material. Thank you again for your valuable advice. We hope our revised manuscript can be accepted for publication.
Reviewer 3 Report
Xiaoyan Wei et al. described the metabolomic characteristics of forage Coix seed (DHS) and edibed Coix 266 seed (XBK) and analyzde the fatty acids that may contribute to the difference in lipid contents in their manuscript. The research were designed appropiate, methods were described suefficient, results were clearly presents and conclusions were connect with obtained results.
The manuscript will require minor revision:
1. In introduction section the information about active ingredients in Coix seeds is insufficient, needs development
2. The sentences in 279 line are repeat
3. Lines 285, 295 - references need be in text, not header
Author Response
Response to the comments of reviewer 3
Xiaoyan Wei et al. described the metabolomic characteristics of forage Coix seed (DHS) and edibed Coix seed (XBK) and analyzde the fatty acids that may contribute to the difference in lipid contents in their manuscript. The research were designed appropiate, methods were described suefficient, results were clearly presents and conclusions were connect with obtained results.
Resopnse: Thank you for your decision and constructive comments on my manuscript. We have carefully considered all comments from the reviewers and revised our manuscript accordingly. The main corrections in the paper and the responds to the comments are as flowing:
Major points:
Comment 1:In introduction section the information about active ingredients in Coix seeds is insufficient, needs development.
Resopnse1: We are very grateful to you for giving us the opportunity to revise our manuscript. As the reviewer's suggestion, we have made the correction at Line 55-57 in the revised manuscript.
Comment 2: The sentences in 279 line are repeat
Resopnse2: Thank you very much for your suggestion. We speculative the reviewer refers to “qualitatively and quantitatively analyzed by LC-QTOF-MS” at Line 276 (original 279) in revised manuscript are repeat. We reconfirmed the sentences again and no repetitions were found. Thank you very much for pointing out it further if our response was not accurate.
Comment 3: Lines 285, 295 - references need be in text, not header
Resopnse3: Thank you for pointing out this problem in our manuscript. we have made correction according to the reviewer’s suggestion at Line 365 and 367.
Round 2
Reviewer 1 Report
the article has been corrected